

# Resistance mechanisms of cereal plants and rhizosphere soil microbial communities to chromium stress

Pengyu Zhao[1,2], Yujing Li[1], Xue Bai[1], Xiuqing Jing[1], Dongao Huo[3], Xiaodong Zhao[1,2], Yuqin Ding[1] and Yuxuan Shi[4]

[1] College of Biological Sciences and Technology, Taiyuan Normal University, Taiyuan, China
[2] Shanxi Key Laboratory of Earth Surface Processes and Resource Ecology Security in Fenhe River Basin, Taiyuan Normal University, Taiyuan, China
[3] Research Center for Plant Resources and Nutritional Health, Taiyuan Normal University, Taiyuan, China
[4] College of Environmental Science and Engineering, Nankai University, Tianjin, China

Corresponding author
Xue Bai, 15235384798@163.com

## ABSTRACT

Agricultural soils contaminated with heavy metals poison crops and disturb the normal functioning of rhizosphere microbial communities. Different crops and rhizosphere microbial communities exhibit different heavy metal resistance mechanisms. Here, indoor pot studies were used to assess the mechanisms of grain and soil rhizosphere microbial communities on chromium (Cr) stress. Millet grain variety 'Jingu 21' (*Setaria italica*) and soil samples were collected prior to control (CK), 6 hours after (Cr_6h), and 6 days following (Cr_6d) Cr stress. Transcriptomic analysis, high-throughput sequencing and quantitative polymerase chain reaction (qPCR) were used for sample determination and data analysis. Cr stress inhibited the expression of genes related to cell division, and photosynthesis in grain plants while stimulating the expression of genes related to DNA replication and repair, in addition to plant defense systems resist Cr stress. In response to chromium stress, rhizosphere soil bacterial and fungal community compositions and diversity changed significantly ($p < 0.05$). Both bacterial and fungal co-occurrence networks primarily comprised positively correlated edges that would serve to increase community stability. However, bacterial community networks were larger than fungal community networks and were more tightly connected and less modular than fungal networks. The abundances of C/N functional genes exhibited increasing trends with increased Cr exposure. Overall, these results suggest that Cr stress primarily prevented cereal seedlings from completing photosynthesis, cell division, and proliferation while simultaneously triggering plant defense mechanisms to resist the toxic effects of Cr. Soil bacterial and fungal populations exhibited diverse response traits, community-assembly mechanisms, and increased expression of functional genes related to carbon and nitrogen cycling, all of which are likely related to microbial survival during Cr stress. This study provides new insights into resistance mechanisms, microbial community structures, and mechanisms of C/N functional genes responses in cereal plants to heavy metal contaminated agricultural soils. Portions of this text were previously published as part of a preprint (https://www.researchsquare.com/article/rs-2891904/v1).

# INTRODUCTION

Heavy metals (HMs) exist as trace elements in natural environments and are mostly found in soils as free metal ions, soluble metal complexes, and exchangeable metal ions (*Ren et al., 2022*). Tannery, metallurgy, electroplating, and anthropogenic activities have led to increases in pollution by the HM chromium (Cr) in environments over the past few decades (*Kayode et al., 2022*). Cr exists in several valence states in environments, although trivalent Cr (III) and hexavalent Cr (VI) are the most stable. Cr (III) is less mobile and less toxic than Cr (VI) while also typically mixing with organic matter components in soils and waters (*Gao et al., 2021*). Cr (VI) is more mobile, toxic, and soluble than Cr (III) and is also over 100 times more toxic than Cr (III) (*GracePavithra et al., 2019*). Cr (VI) has been classified as a class A human carcinogen because of its structural similarity to compounds like sulfates, enabling it to readily enter cells through cell membrane carriers, where it can alter genetic material and act as an oxidant, being reduced to Cr (III) that can then produce harmful free radicals (*DeOliveira et al., 2016*). Agricultural soil microbial populations, crop productivity, and public health are all consequently seriously threatened by the buildup, persistence, and toxicity of Cr in food chains. Nevertheless, Cr has received relatively less attention from researchers due to its complex electrochemical characteristics relative to HM contamination caused by cadmium (Cd) and lead (Pb).

The primary cause of the influence of Cr on processes like plant activation and uptake is its metallic valence state. The toxic effects of Cr primarily inhibit crop yields, root and leaf growth, and enzyme activities. Numerous studies on plant phenotypes have revealed that Cr suppresses root growth in onions (*Allium cepa*) (*Patnaik, Achary & Panda, 2013*); germination and root growth in *Arabidopsis* (*Arabidopsis thaliana*), resulting in leaf yellowing (*Castro et al., 2007*; *Eleftheriou et al., 2015*); biomass, root length, and branching processes in farmed rice plants (*Oryza sativa*) (*Wakeel et al., 2021*); and plant height, root length, and number of tillers in wheat (*Triticum aestivum*) (*Adrees et al., 2015*). Among intrinsic mechanisms, Cr stress modifies catalase (CAT) activity in maize (*Zea mays*) (*Anjum et al., 2017*), inhibits the production of metabolites like auxin (IAA) in plants, and induces oxidative stress (*Fan et al., 2020*), in addition to interfering with plant nutrient absorption, neogenic metabolism, and ultimately stalling plant development and organ abscission. However, plants have various HM detoxification mechanisms, including chelating HMs with molecules like phytochelatins (PCs), metallothioneins (MTs), and glutathione (GSH) to protect plant cells (*Srivastava et al., 2021*). Further, plants can transport HMs to specific tissues with low metabolic activities through vesicular sequestration capacity (VSC), with efficient distribution capacity being an effective HM detoxification mechanism (*Peng & Gong, 2014*). Moreover, the reduction of Cr reduces its toxicity while preventing intracellular oxygenative bursts (OBs) and cellular dysfunction (*Wakeel & Xu, 2020*). In addition, elevated expression of antioxidant enzymes during OBs helps mitigate the negative effects of oxygen anions (*Ayyaz et al., 2021*). Finally, in Cr

stress environments, plants modify signal transduction pathways for hormones like IAA and abscisic acid (ABA) to boost their defenses, collect resources from soils, and control interactions between root systems and soil microbial populations (*Jiménez-Vázquez et al., 2020*). Thus, plant resistance to Cr stress involves changes in several processes, including hormone signal transduction pathways, antioxidant and chelator production, distribution within the plant, and altering accumulation within the organism. Nevertheless, HM contamination of agricultural soils reduces crop quality and poses a serious threat to human food security. Consequently, developing an understanding of crop resistance to Cr stress is critically needed.

Bacterial and fungal communities comprising a vast majority of soil microorganisms that act as important regulators of soil organic matter and nutrient cycling (*Chen et al., 2014*). The structure and function of microbial communities in soils have been extensively investigated as reliable indicators of soil quality and ecological processes, due to their sensitivity to HM pollution (*Wu et al., 2017*). For example, high concentrations of Cu contamination impact the structural composition and α-diversity of soil bacterial communities. Moreover, bacterial strains isolated from contaminated soils often have considerable Cr resistance and Cr reduction capacity (*Zhang et al., 2016*). Further, fungal communities can adapt to soils that have been polluted with Cr for a long period of time (*Liu et al., 2019b*). Nevertheless, environmental factors in tailing wastewaters contaminated with complex HMs have been shown to heavily impact fungal communities (*Liu, Chai & Luo, 2021*; *Liang et al., 2018*). Indeed, numerous studies have demonstrated that HM pollution significantly alters the composition and diversity of soil microbial communities. However, bacterial and fungal communities in soils react differently to different types of stresses and the impacts of stress are correlated with HM concentrations, species, and exposure period.

As suggested above, soil nitrogen and carbon biogeochemical cycling is crucial for the survival of organisms. Microorganisms play key roles in these cycles, including carbon fixation, nitrification, denitrification and so on. Indeed, soil carbon and nitrogen cycles are cooperatively driven and regulated by diverse microbial functional groups that critically maintain ecosystem stability and function (*Bardgett, Freeman & Ostle, 2008*). The structure, diversity, and spatial distribution of microbial populations are directly influenced by community assembly processes, and these changes can indicate habitat alteration (*Liu, Chai & Luo, 2021*). The HM stress resistance mechanisms of microbial communities and the level of soil pollution can consequently be better understood by investigating the structure, diversity, and community-assembly characteristics of soil microbial communities in response to HM stress.

In this study, cereals and rhizosphere microbial communities of Cr-contaminated soils were investigated to examine (1) the resistance mechanisms of cereals to Cr HM stress, (2) the composition, distribution, and assembly mechanisms of soil bacterial and fungal communities in Cr-contaminated soils, and (3) the dynamic changes of functional genes related to carbon and nitrogen cycling in Cr-contaminated soils. Overall, the study aimed to clarify the mechanisms of plant and microbial community stress resistance

in Cr-contaminated agricultural soils, provide new insights into the diversity of HM-contaminated soil microbial communities, and establish a theoretical framework for ecological restoration.

## MATERIALS & METHODS

### Materials and experimental design

The newly developed foxtail millet grain variety 'Jingu 21' (*Setaria italica*) by the Shanxi Academy of Agricultural Sciences and the Industrial Crop Institute was investigated as a model grain variety. The soil used for planting was grass charcoal substrate soil (for seedings). According to research (*Sharma et al., 2020*), the pollution agent was 1 mmol·$L^{-1}$ potassium dichromate ($Cr^{6+}$, $K_2Cr_2O_7$).

Pot trials were conducted in a group culture room (23 °C, White light, 3000 Lx) using alternating 16/8 h light/dark cycles, and the relative humidity was 60%. As a pretreatment for planting, a selection of healthy and plump seeds of 'Jingu 21' with consistent grain size were rinsed and then submerged in water for 48 h. To prepare soils for pot tests, they were air-dried in the lab for 2–3 days. A total of 18 pots (15 cm × 20 cm) were used, with soil humidity maintained at 60% of the water-holding capacity. A total of 1,300 g of soil and 50–60 uniformly spaced seeds were added to each pot. After planting seeds, the soils were mulched to preserve soil moisture and shield seeds from light. The soils were then wrapped in plastic to prevent airborne bacteria from contaminating the seedlings. The pots were then immediately placed in a group culture room at a temperature of 23 °C to encourage germination and grain growth. After 15–20 days of incubation until grain growth was uniform, 300 ml of one mmol · $L^{-1}$ $Cr^{6+}$ was applied to each pot in the test group.

### Sampling plants and soils

According to the research of *Leng et al. (2020)*, soil microbial communities respond to heavy metal stress in around 6h but progressively reach a stable state when heavy metal stress is present for around 6 days, due to resistance and resilience. Based upon this, this experiment was divided into three groups: the CK (addition of equal amounts of distilled water), Cr stress for 6 hours (Cr_6h), and Cr stress for 6 days (Cr_6d).

Each set of soil samples included six replicates for a total of 18 samples. Further, each group of plant samples included three replicates, totaling nine samples. To preserve the integrity of the rhizosphere soils as much as possible, soil samples were extracted and then stored in the refrigerator at −80 °C for subsequent DNA extraction. Fresh plant samples (leaves) were partially flash-frozen in liquid nitrogen and stored at −80 °C for later transcriptomic analysis. Fresh plant leaves were also used for investigation of plant physiological indicators and biomass determination.

### Measurements and analysis

#### Plant physiological indicators and biomass measurements

A Beijing Zhongke Weihe chlorophyll meter (TYS-3N; Zhongke Weihe, Haidian District, Beijing, China) consisting of a pressure-head sensor that emits red and infrared light with peak wavelengths of 650 mm and 940 mm, respectively, was used to measure chlorophyll

and nitrogen contents of seedling leaves, respectively. The working principle is that two LED light sources emit red light (650 nm) and infrared light (940 nm). These two lights penetrate the blade and hit the receiver, converting the light signal into an analog signal. The analog signal is amplified by the amplifier and converted into a digital signal. The digital signal is processed by the processor to calculate the corresponding value, which is displayed on the screen. Three fresh seedling samples were used from each pot and the means of absorbances were calculated. Stem length, root length, and fresh weight of the whole plant were collected for the samples, followed by drying at 105 °C for 30 min and then drying at 60 °C until constant weight to measure dry weight.

### Plant leaf RNA extraction and transcriptome sequence analysis

The statistical power of this experimental design, calculated in RNASeqPower is 0.82. Seedling leaf samples were extracted using an RNAprep Pure Plant Total RNA Extraction Kit (Kaitai Biotechnology Co., Hangzhou, China), purified with a Plant RNA Purification Reagent, and sent to Shanghai Meiji Biotechnology Engineering Co. Ltd. for transcriptomic analysis. RNA sequences were aligned to the refence genome of *Setaria italica* using HISAT2 (version 2.1.0, http://ccb.jhu.edu/software/hisat2/index.shtml) and the BWT algorithm. RSeQC (version 2.3.6) was used to assess the sequencing quality. Quality control of sequencing data was conducted using the fastx_toolkit (version 0.0.14, http://hannonlab.cshl.edu/fastx_toolkit/) by removing spliced sequences from the original reads, trimming low quality bases from the ends of sequences (quality value < 20), and removing N-containing reads to obtain clean reads. Transcript assembly was conducted using StringTie (version 2.1.2, https://ccb.jhu.edu/software/stringtie/).

RSEM (version 1.3.3, http://deweylab.biostat.wisc.edu/rsem/) software was used to quantitatively analyze the expression levels of genes. Expression differences between genes were visually observed by homogenizing gene length and sequencing depth to ensure that the total expression was consistent among different samples. The criteria of $p < 0.01$ and $|log2FC| \geq 2$ were used to identify differentially expressed genes (DEGs) by comparison of Cr stress conditions (Cr_6h and Cr_6d) against CK by multiple checks and corrections using the DESeq2 (version 1.24.0, http://bioconductor.org/packages/stats/bioc/DESeq2/) software program. The BLAST (version 2.9.0) software program was used to compare genes against the Gene Ontology (GO) and Kyoto Encyclopedia of Genes and Genomes (KEGG) databases to assess gene functions. GO annotation analysis can be divided into three major categories according to function: molecular_function (MF), cellular_component (CC), and biological_process (BP). GO and KEGG enrichment analyses were performed to identify the specific functional categories regulated in plant seedlings.

### DNA extraction of soil bacterial and fungal community and high-throughput sequence analysis

A TIANamp Bacteria DNA kit (Tiangen Biochemical Technology Co., Beijing, China) was used to isolate DNA from 1 g of each soil sample. The purity and concentrations of the extracts were assessed using agarose gel electrophoresis and A260/A280 absorbances, respectively, with values of the latter between 1.8 and 2.0 used to identify high purity extracts. Quantitative polymerase chain reaction (PCR) was used to amplify 16S rRNA

(bacterial) or 18S rRNA (fungal) gene fragments from genomic DNA. Universal bacterial and fungal primers used for PCR were 338F_806R (338F: ACTCCTACGGGAGGCAGCAG; 806R: GGACTACHVGGGTWTCTAAT) and ITS1F_ITS2R (ITS1F: CTTGGTCATT-TAGAGGAAGTAA; ITS2R: GCTGCGTTCTTCATCGATGC) (*Caporaso et al., 2011*), respectively. The PCR amplification systems comprised a total volume of 50 μL: 1.0 μL Taq DNA polymerase (5 U · μL$^{-1}$), 10.0 μL 10 × PCR buffer, 8.0 μL dNTP solution, 10.0 μL template DNA, and primers (50 μmol · L$^{-1}$). PCR reaction conditions included pre-denaturation at 98 °C for 1 min, followed by 30 cycles of 98 °C for 10 s, 50 °C for 30 s, and 72 °C for 30 s, with a final extension at 72 °C for 5 min (*Zeng & An, 2021*). The PCR amplicon samples were sent to Shanghai Meiji Biotechnology Engineering Co., Ltd. for high-throughput sequencing on the Illumina Miseq sequencing platform.

High-quality amplicon sequences were obtained by paired-end sequencing and then spliced using the Flash software program (version 1.2.11, https://ccb.jhu.edu/software/FLASH/index.shtml), followed by quality control with the Fastp program (version 0.19.6, https://github.com/OpenGene/fastp). Sequences were clustered into operational taxonomic units (OTUs) using the Uparse software program (version 7.0.1090, 7.0.1090, http://drive5.com/uparse/) at a 97% nucleotide sequence similarity level. The RDP classifier Bayesian algorithm was used for taxonomic classification of OTU representative sequences, and species annotation analysis was conducted based on OTU abundance data to enumerate the compositions of bacterial and fungal communities in the soils. A table relating the taxonomic abundances and β-diversity distances was constructed using the QIIME software program (version 1.9.1, http://qiime.org/scripts/assign_taxonomy.html) software. In addition, α-diversity analysis (community richness and diversity index calculations) was performed using the Mothur software program (version 1.30.2, https://www.mothur.org/wiki/Download_mothur).

### Real-time fluorescence quantitative PCR (qPCR)

Seven microbial functional genes were quantified using qPCR, including those encoding the archaeal ammonia monooxygenase (AOA-*amoA*), bacterial ammonia monooxygenase (AOB-*amoA*), nitrate reductase (*narG*), nitrite reductase (*nirK*), nitrogen-fixing enzyme (*nifH*), methane monooxygenase (*pmoA*), and methyl coenzyme M reductase (*mcrA*). The primers used to amplify the functional genes are shown in Table S1 (*Hu et al., 2022*).

An ABI Model 7300 Fluorescent Quantitative PCR machine (Applied Biosystems, Foster City, CA, USA) and ChamQ SYBR Color qPCR Master Mix (2X) reagent (Nanjing Novozymes Biotechnology Co., Ltd.) were used for qPCR. The final reaction mixture volume for quantification was 20 μL, which contained 10 μL 2X Taq Plus Master Mix, 1 μL template DNA, 0.8 μL forward primer (5 μM), 8 μL reverse primer (5 μM), and 7.4 μL ddH$_2$O. Each qPCR was conducted in triplicate for each gene, and a non-template control was used as the negative control. qPCR was conducted using a three-step thermal cycling method that included pre-denaturation at 95 °C for 5 min, 35 cycles of denaturation at 95 °C for 30 s, annealing at 58 °C for 30 s, and an extension at 72 °C for 1 min. (*Li et al., 2022*). To create standard curves, the plasmid pMD18-T was used with seven functional genes and 10-fold dilution gradients. The R$^2$ value for each standard curve was >0.95, demonstrating

a linear relationship over the study's concentration range, with amplification effectiveness ranging from 89.28% to 103.81% (mean 97.53%).

## Data analysis

Data processing and visualization were conducted in Excel 2010 and R (version 4.1.2) software programs. Changes in plant physiological indicators and biomass, in addition to soil microbial diversity and assembly mechanisms, were evaluated between different treatment groups of Cr stress using one-way analysis of variance (ANOVA) tests. Post hoc comparisons were conducted by the Turkey's test. The above analysis is implemented using the glht function in the 'multcomp' package in R. Volcano plots were used to identify up- and down-regulated plant DEGs among Cr stress treatment groups. GO and KEGG functional annotation and enrichment analyses were used to identify differences in the functional expression of DEGs. The Sobs and Shannon indices, in addition to other *alpha* diversity metrics of soil microbial communities, were investigated. Non-metric multidimensional scaling analysis (NMDS) was used to assess the geographical and temporal distribution patterns of soil microbial communities. Analytical results were confirmed by stress values; when the stress value was less than 0.2, ordinations were considered to have explanatory importance.

Community network analysis was conducted using the igraph package for R (4.1.2) and visualized using the Gephi software program (https://gephi.org/). Phylogenetic structure-based analysis, as proposed by *Stegen et al. (2013)*, was used to determine the drivers of soil community assembly. When the median *beta* NTI values across all samples were $> -2$ and $<2$, stochastic processes were inferred to dominate the construction of the microbial communities. When the median *beta* NTI values were $>2$ or $<-2$, deterministic mechanisms were inferred to have structured the communities.

## RESULTS

### Growth analysis of millet seedlings

Plant stem lengths, root lengths, dry weights, fresh weights, chlorophyll contents, and nitrogen contents were evaluated to assess how Cr stress affected grain growth (Table 1). The results of one-way ANOVA showed that, compared with CK, the chlorophyll and N content of cereals decreased significantly by 23.66% and 17.36% at Cr_6h stage ($p < 0.05$). Compared with Cr_6h, the stem length, dry weight, and fresh weight of cereals increased significantly by 40.04%, 47.06% and 28.57% at Cr_6d stage, respectively ($p < 0.05$).

### Transcriptome analysis of cereal seedlings in Cr-stressed soils
#### Transcriptome sequencing quality assessment

Transcriptome sequencing produced 57.19 Gbp of clean data, with the Q20, Q30, and GC statistics of each sample's clean reads being 98.20%, 94.60%, and 53.80%, respectively (Table S2). Thus, the sequencing data were satisfactory and appropriate for further data analysis. *Setaria italica* was used as the reference genome, and comparison to the genome yielded a 93.93%–95.71% identification rate. Thus, the comparison results indicated that they were sufficient for further annotation and analysis.

**Table 1 Determination of millet biomass and physicochemical properties in the Cr stress time series (mean ± SE).** Data are mean ± standard deviation, and different lowercase letters indicate significant differences between components ($p < 0.05$).

| Treatment group | Control (CK) | Chromium stress for 6 h (Cr_6h) | Chromium stress for 6 days (Cr_6d) |
|---|---|---|---|
| Stem length (cm) | 5.86 ± 0.55b | 5.52 ± 0.55b | 7.73 ± 1.12a |
| Root length (cm) | 4.18 ± 0.81a | 3.64 ± 0.23a | 3.83 ± 0.75a |
| Dry weight (g) | 0.0031 ± 0.0005b | 0.0034 ± 0.0005b | 0.005 ± 0.0006a |
| Fresh weight (g) | 0.0283 ± 0.0053b | 0.0294 ± 0.0026b | 0.0378 ± 0.0051a |
| Chlorophyll (SPAD) | 24.68 ± 1.70a | 18.84 ± 1.77b | 16.34 ± 2.00b |
| N (mg/g) | 7.49 ± 0.51a | 6.19 ± 0.52b | 5.45 ± 0.58b |

## Identification of DEGs

TPM algorithm-based calculations of negative binomial distributions were used to identify DEGs between groups, based on the criteria of $p < 0.01$ and $|\log_2 \text{Foldchange}| \geq 2$. The DEG expression levels were visualized with volcano plots (Figs. 1A, 1C, 1D). The Cr_6h group contained 3,906 DEGs (2,483 upregulated and 1,423 downregulated) compared to the CK group, while the Cr_6d group contained 4,382 DEGs (1,927 upregulated and 2,455 downregulated) compared with the CK group. The Cr_6h group contained 2,090 DEGs (1,156 upregulated and 934 downregulated) compared to the Cr_6d group. Thus, temporal heterogeneity in gene expression was observed for cereal plants after Cr stress, with distinct DEGs present at different times. Gene expression patterns in grains at Cr_6h and Cr_6d were more distinct, illustrating the specificity underlying grain responses to Cr stress. Increased Cr stress time was associated with increased DEGs in the Cr_6d group. Up- and down-regulated DEGs comprised roughly 54% and 46% of the total genes, respectively, suggesting that Cr stress increased grain gene expression. Additionally, only 8.90% of the genes were shared by the three groups of samples (Fig. 1B), meaning that only 8.90% of the genes in cereal seedlings consistently responded to Cr stress.

## GO functional annotation and enrichment analysis of DEGs

The identified DEGs were subjected to GO functional annotation analysis to assess the primary biological functions associated with DEGs in cereals at different time points of Cr stress (Fig. 2). The results indicated that the significantly enriched items of DEGs in the three groups of sample pairs included catalytic activity, transporter protein activity, and transcription regulator activity in MF, cell part and membrane part in CC, as well as metabolic process, cellular process, and others in BP.

The upregulated DEGs for the CK & Cr_6h group (Fig. S1) were significantly enriched in the categories of DNA replication licensing factor-MCM complex, DNA replication, and pre-replicative complex assembly. For the CK and Cr_6d comparison, DEGs were enriched in the categories of exosome (RNase complexe) and cellular aromatic compound metabolic. For the Cr_6h and Cr_6d groups, the categories of jasmonic acid mediated signaling pathway, regulation of defense response, and kinase activity and transferase activity were particularly enriched. The downregulated DEGs in CK and Cr_6h comparison were significantly enriched in the categories of photosynthetic light reaction processes,

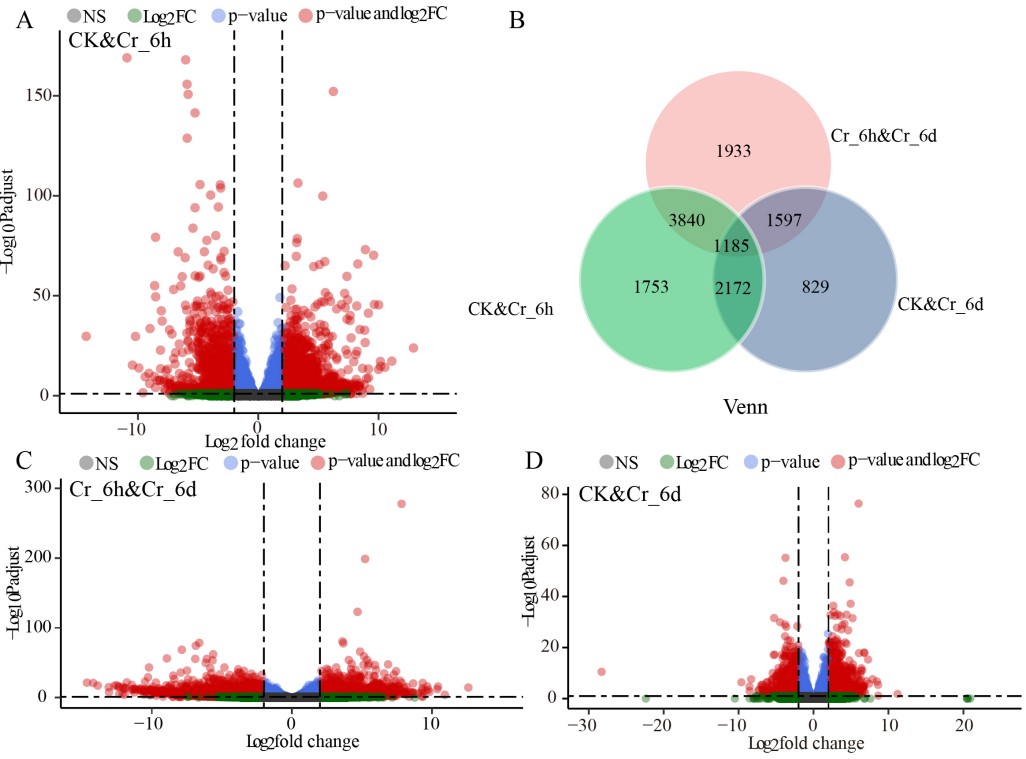

**Figure 1 Differentially expressed genes (DEGs) in leaves of *Setaria italica* in the Cr stress time series.**
(A, C, D) Volcano plots of DEGs in leaves of *S. italica* from three different treatments; X-axis indicates log₂ fold change, Y-axis indicates −log₁₀P adjust; NS: not significant; log₂FC: −log₁₀Padjust <5 and log₂ fold change absolute value > 2; p-value: −log₁₀Padjust >5 and log₂ fold change absolute value <2; p-value and log₂FC: −log₁₀Padjust >5 and log₂fold change absolute value >2. (B) Venn diagram showing the effect of different samples on DEGs in leaves of *S. italica*.

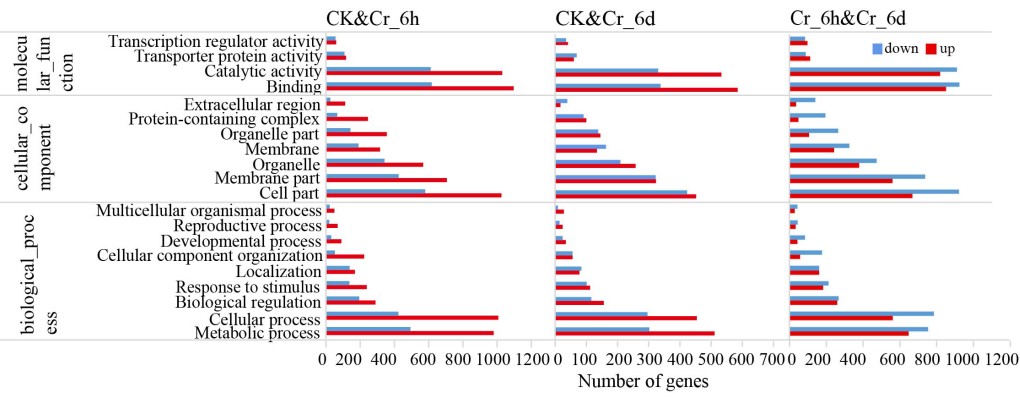

**Figure 2 GO annotation analysis of DEGs in leaves of *S. italica* under Cr treatment.**

oxidoreductase activity, and responses to abiotic stimulus. The CK and Cr_6d group downregulated DEGs were significantly enriched in the categories of photosynthesis,

light reaction, and photosynthesis, dark reaction and photosynthetic electron transport chain. Finally, the downregulated DEGs of the Cr_6h and Cr_6d groups were significantly enriched in categories related to microtubule-based movement, cell wall organization or biogenesis, and subcellular component. Overall, the expression of genes involved in photosynthesis and cell membranes was significantly reduced in cereal leaves due to Cr stress.

### KEGG functional annotation and enrichment analysis of DEGs

The KEGG functional annotations of DEGs were evaluated to investigate the metabolic regulatory processes and signal transduction pathways responsive to Cr exposure stress in cereals. DEGs were considerably enriched in the three sample comparisons for the KEGG pathways of phenylpropanoid biosynthesis, carbon fixation in photosynthetic organisms, and amino acid biosynthesis and metabolism (Fig. S2).

Upregulated DEGs in the CK and Cr_6h comparison (Fig. 3) were significantly enriched in the KEGG pathways of DNA replication; cutin, suberine, and wax biosynthesis; and starch and sucrose metabolism. Upregulated DEGs of the CK and Cr_6d comparison were significantly enriched in categories of ribosome biogenesis in eukaryotes and plant-pathogen interaction. In addition, the upregulated DEGs of the Cr_6h and Cr_6d comparison were significantly enriched for the categories of the MAPK signaling pathway, plant hormone signal transduction, and glycolysis/gluconeogenesis processes. Downregulated DEGs of the CK&Cr_6h group were significantly enriched in pathways related to carbon fixation in photosynthetic organisms and amino acid metabolism and nitrogen metabolism. Downregulated DEGs of the CK and Cr_6d comparison were significantly enriched in the photosynthesis-antenna proteins and oxidative phosphorylation category, while those of the Cr_6h and Cr_6d comparison were significantly enriched in the categories of phenylpropanoid biosynthesis and plant hormone signal transduction (Fig. 3).

Investigation of the Clusters of Orthologous Group (COG, Table S3) functional annotations revealed that the COG functions of DEGs after Cr stress were primarily associated with carbohydrate transport and metabolism; signal transduction mechanisms; amino acid transport and metabolism; and intracellular trafficking, secretion, and vesicular transport.

### DEG expression profile clustering

The Short Time-series Expression Miner program (STEM, version 1.3.11) was used to cluster the DEG profiles and visualize the temporal dynamics of gene expression (Fig. 4A). The DEGs were divided into 16 clusters, with four exhibiting significant clustered gene expression patterns ($p < 0.05$), including two distinct cluster groupings. Comparison of Cr_6h to Cr_6d revealed that the DEGs in Profiles 10 and 14 exhibited a clear upward trend, followed by a downward trend. In Profile 11, the DEGs exhibited a clear increasing trend at Cr_6h, followed by a stable trend from Cr_6h to Cr_6d. The DEGs in Profile 15 exhibited a substantial increasing trend at Cr_6h and Cr_6d. The TCseq package for R was then used to cluster the DEGs of the four significant STEM modules using fuzzy c-means clustering to further validate their groupings. The vertical coordinate Z-score was used

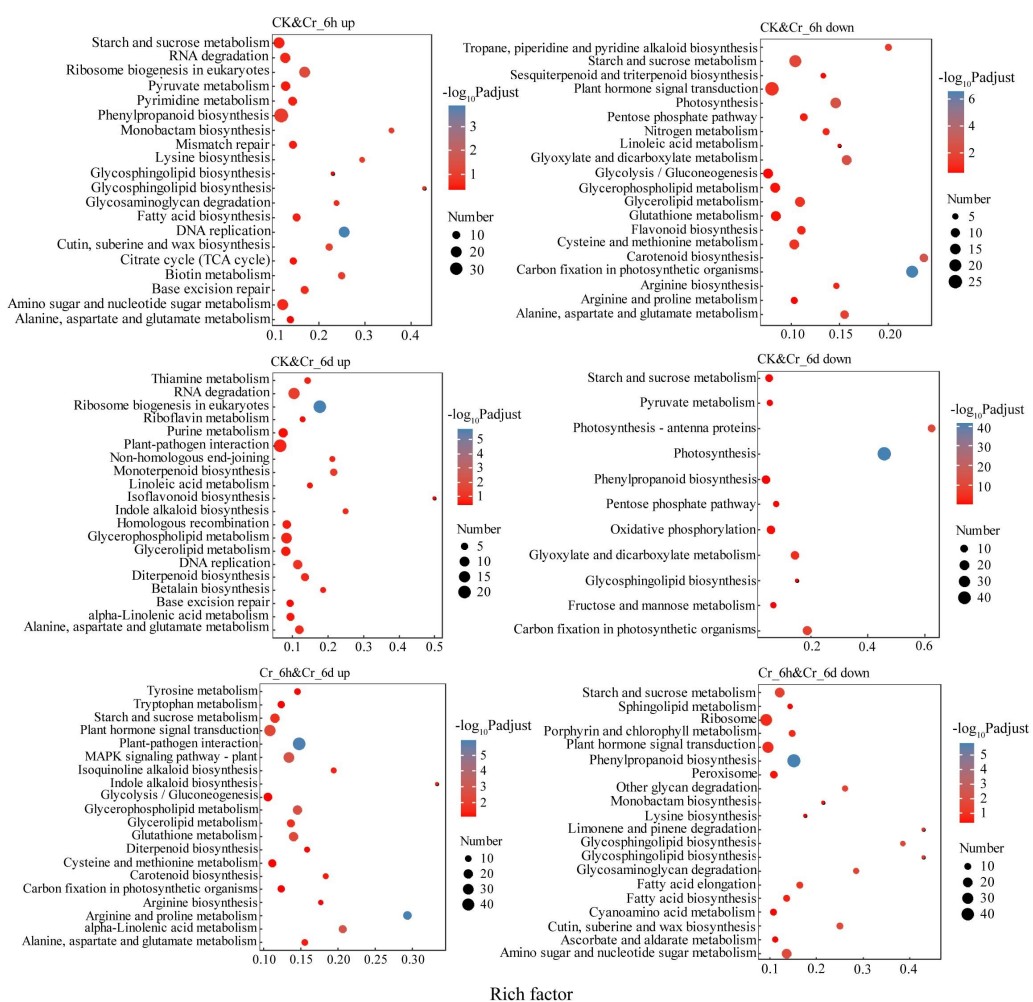

**Figure 3  KEGG enrichment analysis of DEGs in Cr stressed *S. italica* leaves.**

to ensure that the standardized variables comprised comparable data, and the clustering groups to which the DEGs belonged were assessed by membership values (Fig. 4B). The four significant clusters were consistent with those identified in the STEM analysis.

GO functional enrichment analysis was further used to identify the functions of the STEM analysis gene clusters with the same temporal expression patterns (Table S4). DEGs in Profiles 10 and 14 were significantly enriched in processes related to intercellular transport,plasmodesmata-mediated intercellular transport, chloroplast fission, plant-type secondary cell wall biogenesis processes, regulation of decapping enzyme activity, and metabolic processing of RNA-related molecules. The DEGs of Profile 11 were significantly enriched in functions related to ribonucleoprotein complex biogenesis, tRNA and rRNA methylation modification, prptidyl-amino acid modification, and other processes. The DEGs of Profile 15 were considerably enriched in processes involved in RNA processing and metabolism, cellular aromatic compound metabolic process, carbohydrate derivative metabolic process, cellular nitrogen compound metabolic process, glucose-6-phosphate

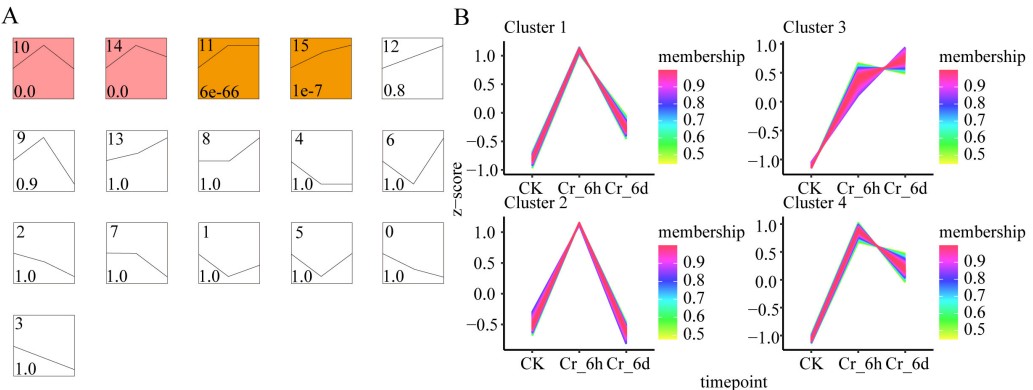

**Figure 4  Gene expression profiles of *S. italica* leaves in Cr-contaminated habitats.** (A) Analysis of the temporal expression trend of DEGs in leaves of *S. italica*. Each profile corresponds to a rectangle, and the number in the upper left corner of the rectangle is the profile number, starting from 0. The broken line in it represents the trend of expression level changing over time, and the value in the lower left corner is the corresponding significance level p values. Trends of clusters with white backgrounds are not significant, while those of clusters with colored backgrounds are significant ($p < 0.05$). Clusters of the same color indicate similar trends in the expression of the genes they comprise. (B) Cluster analysis of gene expression under different periods of Cr stress.

metabolic processes, and intracellular vesicle. These results suggest that fundamental metabolic pathways of plants, including metabolism of carbohydrates, glycans, and nitrogen, are involved in the early responses to Cr stress in cereals. The DEGs in Profile15 exhibited a substantial increasing trend and were significantly enriched in BP functions compared to CC and MFs.

## High-throughput sequence analysis of Cr-stressed rhizosphere soil microbial communities of cereals

### Structural analysis of bacterial and fungal community composition

The five most abundant bacterial phyla in cereal rhizosphere soils (Fig. 5A) were the Actinobacteriota (31.09%), Firmicutes (18.95%), Proteobacteria (18.78%), Chloroflexi (11.95%), and Bacteroidota (4.56%). Compared with CK, the relative abundance of Proteobacteria significantly decreased ($p < 0.05$) at Cr_6d stage. The five most abundant fungal phyla in soils (Fig. 5C) were the Ascomycota (87.64%), Basidiomycota (3.12%), Chytridiomycota (1.56%), Mortierellomycota (1.13%), and Rozellomycota (1.13%). After Cr stress, compared with CK, the relative abundance of Chytridiomycota significantly decreased ($p < 0.05$) at Cr_6h stage, and then showed a significant rise between the Cr_6h and Cr_6d treatments ($p < 0.05$).

The five most abundant bacterial genera (Fig. 5B) were *Planifilum* (6.34%), *Longispora* (4.56%), *Actinomadura* (2.66%), *Haloactinopolyspora* (2.51%), and *Truepera* (2.22%). The relative abundance of *Planifilum* significantly decreased ($p < 0.05$) at Cr_6d stage compared with Cr_6h. The five most abundant fungal genera (Fig. 5D) were *Chaetomium* (8.51%), *Gibberella* (7.26%), *Fusarium* (6.58%), *Chrysosporium* (5.60%), and *Lophotrichus* (4.64%). The relative abundance of *Gibberella* significantly decreased ($p < 0.05$) at Cr_6d

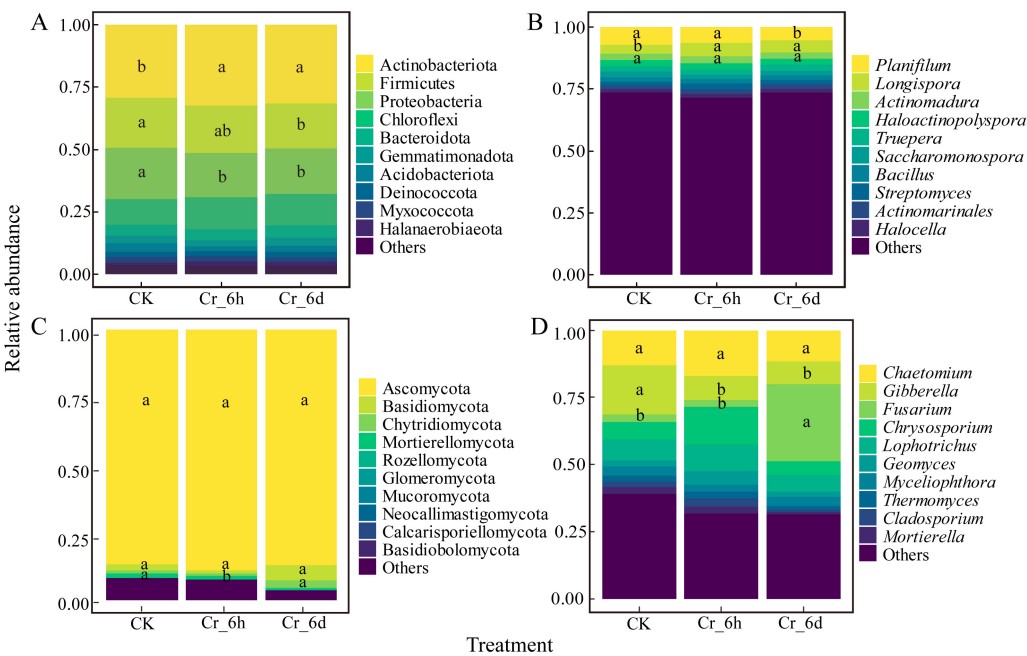

**Figure 5   Bacterial and fungal community compositions at the phylum and genus levels in the Cr stress time series.** (A, C) Community composition at the bacterial and fungal phylum levels, respectively. (B, D) Community composition at the bacterial and fungal genus levels, respectively. 'Others' represents the sum of the relative abundance of the remaining bacteria and fungi. Different lowercase letters indicate statistically significant differences between components. CK, control CK; Cr_6 h, Cr stress for 6 hours; Cr_6d, Cr stress for 6 days.

stage compared with CK, while that of *Fusarium* significantly increased ($p < 0.05$). Thus, soils under Cr stress exhibited considerable changes in the structures and compositions of bacterial and fungal communities.

### α *and* β *diversity of soil bacterial and fungal community*

Changes in the α diversity (Sobs, Shannon, Simpson, ACE, and CHAO indices) of bacterial and fungal communities of soils under Cr stress were determined using ANOVA tests. The Sobs and ACE indices of soil bacterial communities did not significantly change across treatments ($p > 0.05$, Fig. 6A). Compared with CK, the Shannon index decreased significantly by 2.32% at Cr_6h stage, while the Simpson index increased significantly by 14.93% ($p < 0.05$). Compared with Cr_6h, the Shannon index increased significantly by 1.39% at Cr_6d stage, while the Simpson index decreased by 10.39% ($p < 0.05$). Among the fungal community indices, only the Shannon index decreased by 6.56% ($p < 0.05$, Fig. 6B) between the CK and Cr_6d treatments. Overall, the results reveal that under Cr stress, both the fungal and bacterial communities exhibit significant phased change characteristics ($p < 0.05$).

NMDS and analysis of similarities (ANOSIM) tests were used to examine the β-diversity of bacterial and fungal communities in soils after Cr stress. The soil community structures of bacteria (stress = 0.071, $p = 0.001$, Fig. 6C) and fungi (stress = 0.059, $p = 0.001$, Fig. 6D)

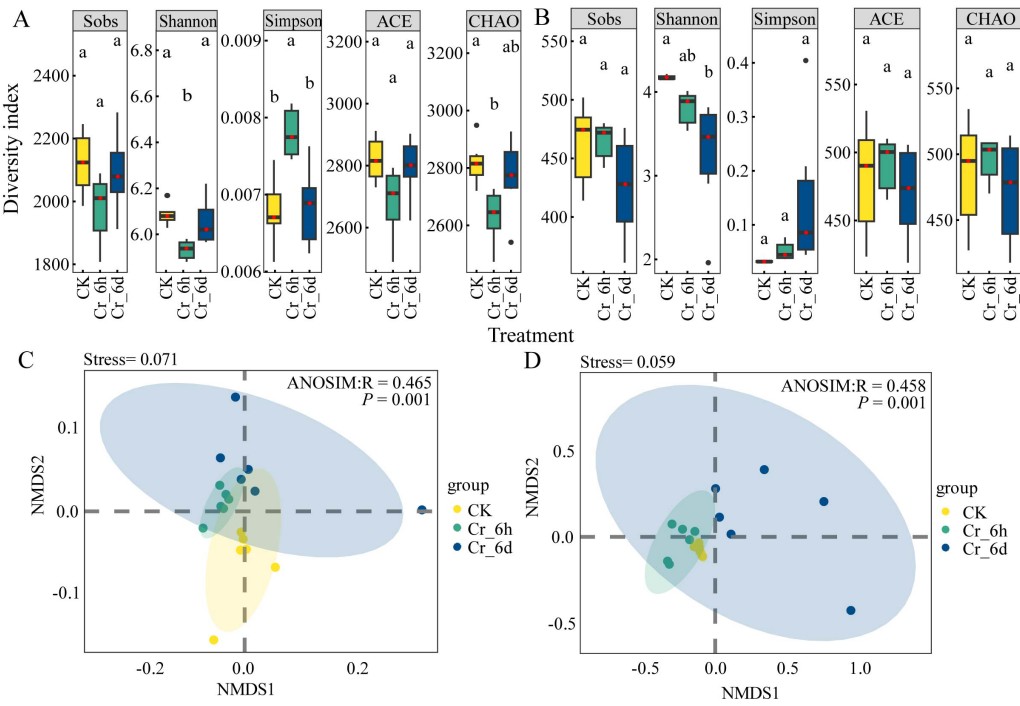

**Figure 6** $\alpha$-diversity and $\beta$-diversity of soil bacterial and fungal communities in the Cr stress time series. (A, B) Analysis of bacterial and fungal community diversity indexes in the Cr stress time series. The black dots represent outliers, and the upper, middle, lower lines of the box chart represent the upper quartile, median, and lower quartile, respectively. The vertical black line is the error bar. (C, D) Non-metric multidimensional scale analysis of bacterial and fungal communities on the Cr stress time series. Points of different colors or shapes represent samples of different treatments. The closer the two sample points, the more similar the species composition of the two samples.

were clustered based on Cr stress stage but were highly different across treatments. Thus, Cr stress led to considerably distinct temporal variation of bacterial and fungal communities.

## Assembly of soil rhizosphere bacterial and fungal communities in Cr-contaminated soils

### Mechanisms of community assembly

The processes of soil microbial community assembly in Cr-stressed soils were investigated based on *beta* NTI values. Deterministic processes (|*beta* NTI|>2) dominated the assembly processes for CK and Cr_6h bacterial communities, while community assembly was dominated by stochastic processes (|*beta* NTI|<2) for Cr_6d communities (Fig. 7A). The median *beta* NTI values for the CK, Cr_6h, and Cr_6d communities were −2.68, −2.11, and −1.91, respectively. In contrast, stochastic processes (|*beta* NTI|<2) dominated the assembly of fungal communities, and the *beta* NTI values for the CK, Cr_6h, and Cr_6d fungal communities were − 0.16, −0.71, and −0.23, respectively (Fig. 7B).

### Microbial co-occurrence network analysis

Co-occurrence network analysis of bacterial and fungal communities was used to assess the correlations in occurrence among different bacterial groups using Spearman

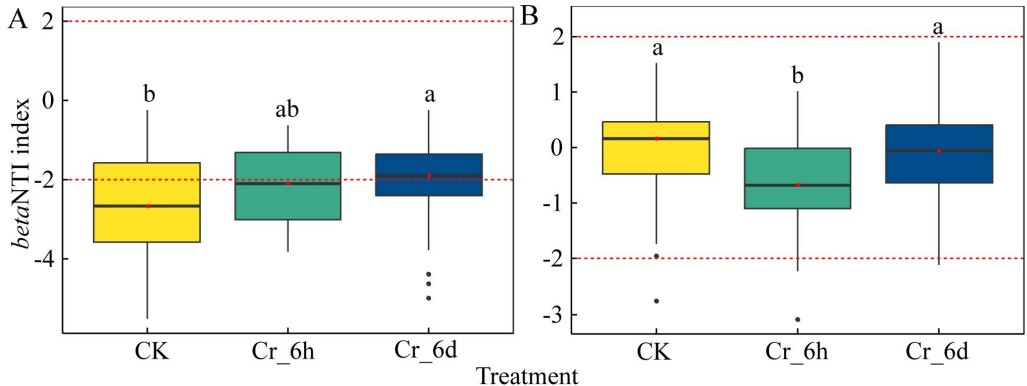

**Figure 7** *Beta* **NTI values of soil bacterial (A) and fungal (B) communities in the Cr stress time series.** The black dots represent outliers, and the upper, middle, lower lines of the box chart represent the upper quartile, median, and lower quartile, respectively. The vertical black line is the error bar, and the red dotted line represents the *beta* NTI value of 2 and −2, respectively.

**Table 2   Soil bacterial and fungal community network attributes following Cr stress.**

| Treatment group | Control (CK) | | Chromium stress for 6 h (Cr_6h) | | Chromium stress for 6 days (Cr_6d) | |
|---|---|---|---|---|---|---|
| | Bacteria | Fungi | Bacteria | Fungi | Bacteria | Fungi |
| Number of edges | 2340 | 17 | 1858 | 17 | 1303 | 36 |
| Number of nodes | 197 | 18 | 195 | 21 | 199 | 21 |
| Average degree | 23.76 | 1.89 | 19.06 | 1.62 | 13.10 | 3.43 |
| Clustering coefficient | 0.57 | 0.71 | 0.52 | 0.5 | 0.57 | 0.47 |
| Modularity | 0.39 | 0.75 | 0.37 | 0.76 | 0.57 | 0.50 |
| Average path length | 2.81 | 1.27 | 3.16 | 1.41 | 3.75 | 3.08 |

correlation analysis of relative abundances. The co-occurrence relationships of soil microbial communities in Cr-stressed soils were then visually explored using co-occurrence network diagrams. Analysis of the bacterial network topological characteristics revealed that the number of nodes, modularity, and average path length were higher in the Cr_6d communities than in the CK and Cr_6h communities, but the numbers of edges and the average degree in the bacterial community continued to decrease with increasing Cr stress time (Table 2). Thus, the bacterial co-occurrence network in Cr_6d soils was larger, the community was more dispersed, modularity increased, and intra-species interactions are inferred to be greater compared to CK and Cr_6h soil. In the fungal network, the numbers of edges, average degree, and average path length of the co-occurrence network were higher for the Cr_6d soils than for the CK and Cr_6h soils, indicating that the network for the former was larger and that interspecific interactions are inferred to be more complicated.

The key bacterial species of community networks have changed under different treatments. The key bacterial species of community networks from different treatment differed (Fig. 8A). Specifically, the key genera in the CK network were *Planifilum*, *Bacillus*, and *Streptomyces*, but in the Cr_6h network, the key genera were *Truepera*, *Actinomadura*,

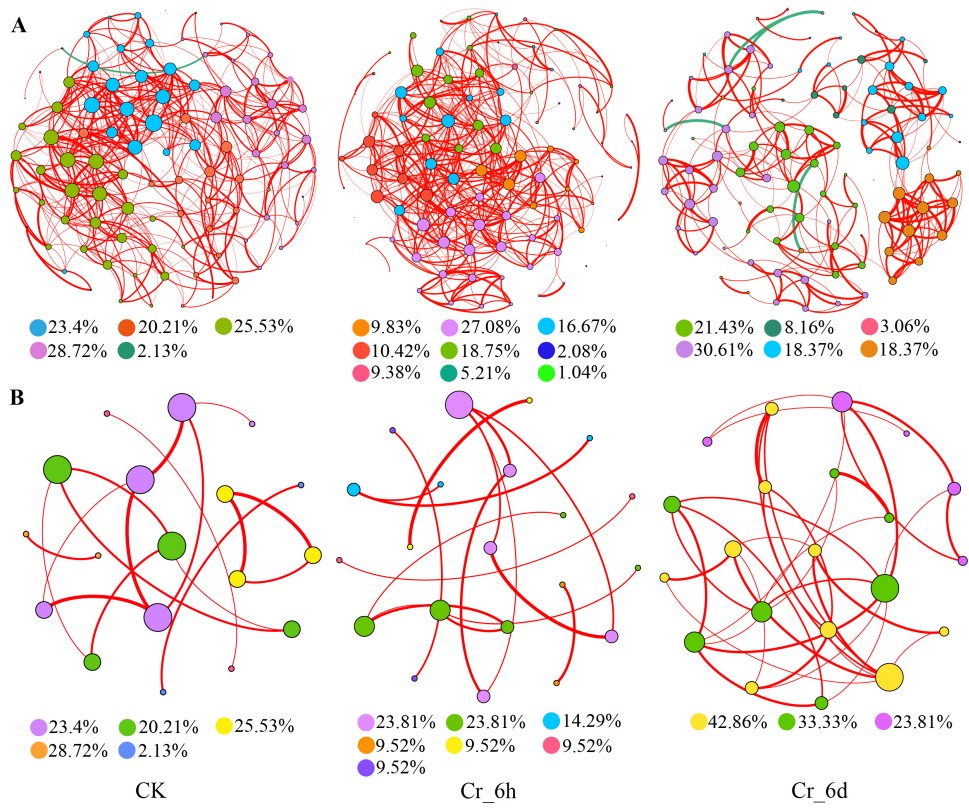

**Figure 8   Interspecies network diagram of bacterial (A) and fungal (B) communities in the Cr stress time series.** Nodes represent different bacterial and fungal genera. Node sizes indicate the degree of centrality for the genus. Larger nodes indicate that they are more important to the network. The red edges indicate statistically significant positive correlations between two nodes. Edge thickness is positively related to correlation significance. Nodes are colored based on the modules they belong to.

and *Arthrobacter*. In addition, the key genera of the Cr_6d network were *Marmoricola* and *Steroidobacter* from the Actinobacteriota and Proteobacteria phyla, respectively. Overall, the dominant genera of the soil bacterial community networks at different stages belonged to the Actinobacteriota and *Proteobacteria* that were likely more resistant to HMs. Bacterial community modules were consistent with the results shown in Table 2, with positive correlations among members in the same module, indicating that the bacterial communities were primarily symbiotic after Cr stress treatment, and their competitive relationships were minimal. Further, the interspecific relationships among community members were characterized by advancement in Cr exposure time.

The key fungi within different soil networks also varied (Fig. 8B). Specifically, *Chrysosporium* and *Chaetomium* were key taxa in the CK soils, while *Gibberella* and *Fusarium* were key taxa in the Cr_6h network. Finally, *Chrysosporium* and *Penicillium* were key taxa of the Cr_6d network. Most of the fungi in the fungal networks were pathogenic, and the distributions of the fungal community modules were consistent with the results shown in Table 2. Positive correlations were dominant within the same module, indicating

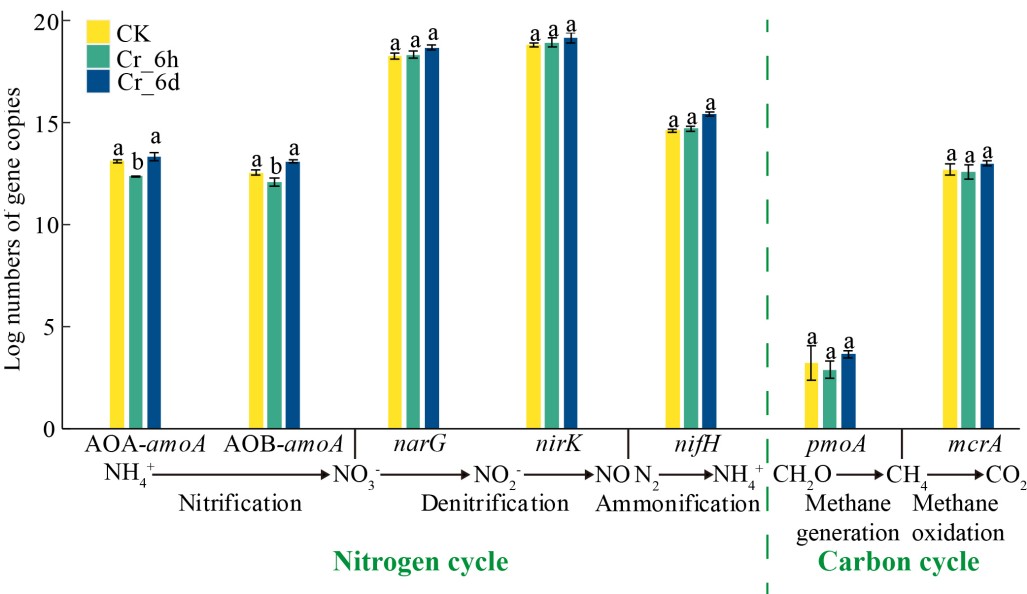

**Figure 9 Abundances of functional genes involved in nitrogen and carbon cycling in Cr-contaminated soils.** Error bars indicate standard errors ($n = 3$), different lowercase letters indicate statistically significant differences between components ($p < 0.05$).

that the fungal communities were dominated by symbiotic relationships following Cr stress and that the competitive relationships were relatively weak.

## Quantitative analysis of microbial functional genes involved in carbon and nitrogen cycling

The expression levels of AOB-*amoA*, AOA-*amoA*, *narG*, *nirK*, *nifH*, *pmoA*, and *mcrA* were evaluated with qPCR to better understand the impacts of Cr stress on carbon and nitrogen cycling gene abundances (Fig. 9). Under Cr stress, compared with the CK soils, the abundances of AOA-*amoA* and AOB-*amoA* decreased significantly by 5.78% and 3.75% ($p < 0.05$), respectively, in the Cr_6h soils, and increased significantly by 1.65% and 4.24% ($p < 0.05$), respectively, in the Cr_6d soils. Thus, the expression of microbial genes involved in nitrogen and carbon cycling exhibited an overall increasing trend in Cr-stressed soil.

## DISCUSSION

### Resistance of cereal seedling leaves to Cr stress

Millet resisted Cr stress by inhibiting the expression of genes related to cell wall, cell membrane, and cell division. The first line of defense for plant resistance to HM stress is *via* cell walls and cytosols, wherein negatively charged functional groups on their surfaces are highly effective at binding HMs to lessen their toxic effects (*Srivastava et al., 2021*). A total of 790 DEGs were involved in the differential regulation of genes related to cell walls, suggesting that cell walls are involved in the responses of cereal plants to Cr stress, consistent with STEM studies. GO enrichment analysis (Fig. S1) revealed that genes related to cell wall composition were considerably downregulated in the Cr_6h and Cr_6d

treatments, and a significant downregulation also occurred in CK and Cr_6d treatments. Further, cutin, suberine, and wax biosynthesis genes were considerably downregulated in the Cr_6h and Cr_6d treatments based on KEGG enrichment analysis (Fig. 3). Cuticles in the above-ground components of plants and the corky substances widely distributed in roots are lipid-phenolic hydrophobic barriers that have evolved to allow plants to adapt to complex and variable external environments. Consequently, reduced wax biosynthesis could lead to reduced wax deposition in leaves, increased stomatal opening, and accelerated leaf shrinkage (*Rai & Mehrotra, 2008*). GO enrichment analysis (Fig. S1) also revealed significant downregulation of the expression of genes involved in microtubule movement and microtubule-related complexes for the Cr_6h and Cr_6d plants, suggesting that Cr stress may prevent leaf mitosis while lengthening cell cycles and preventing leaf cell proliferation and differentiation (*Hossain et al., 2012*). This inhibition restricts nutrient uptake and transport activity in the above-ground components of plants, ultimately resulting in lost leaf biomass of cereal plants. Similar outcomes have been observed for rice roots under Cr (VI) stress (*Sundaramoorthy et al., 2010*). Moreover, COG enrichment analysis (Table S3) revealed that 182 DEGs were enriched in intracellular transport, secretion, and vesicular transport. Vesicular transport mechanisms are involved in plant regulation of environmental stresses, wherein cells can control the effects of external stresses by controlling the rate of transfer between plasma membranes (*Peng & Gong, 2014*).

Millet resisted Cr stress by inhibiting the expression of photosynthesis-related genes in its leaves. Consistently, a significant decrease in chlorophyll content was observed in wheat leaves after increased Cr stress time. Further, GO and KEGG enrichment analyses (Fig. 3, Fig. S1) revealed that genes involved in photosynthesis were significantly enriched in the downregulated genes for all three sample pairs. These results could be due to the capacity of Cr to degrade d-aminolevulinic acid dehydratase, a crucial enzyme involved in chlorophyll biosynthesis, that would then impact how well d-aminolevulinic acid (ALA) is used, resulting in a buildup of ALA and a decrease in chlorophyll levels (*Vajpayee et al., 2001*). Chlorophyll is the primary pigment involved in photosynthesis and its decrease will directly impact how well cereal leaves perform photosynthesis, in turn affecting nutrient uptake, distribution, and transformation within plants and ultimately affecting plant growth. Cr is also known to inhibit photosystems, interfere with electron transport, reduce the activity of the PS II reaction center, and induce PS II heterogeneity (*Mathur, Kalaji & Jajoo, 2016*), consistent with the results of this study (Fig. 3, Fig. S1). Further, Cr (VI) toxicity may cause thylakoid aberrations that adversely impair photosynthesis and lead to imbalances of intracellular energy transfer that would then explain the considerable downregulation of genes related to thylakoid membranes (*Ali et al., 2013*). Thus, Cr stress inhibits cereal growth and development by disrupting photosynthesis in seedling leaves.

In order to resist Cr stress, millet also stimulated the expression of genes involved in DNA replication and repair. Cr toxicity causes organisms to produce reactive oxygen species (ROS) and prevents mitochondrial electron transport in higher plants, thereby causing membrane lipid peroxidation and oxidative stress in plant cells (*Costa et al., 2010*). In this study, genes related to cellular redox homeostasis and oxidoreductase activity were significantly downregulated in the CK and Cr_6h plants, while genes related to

ROS metabolic processes and hydrogen peroxide catabolic processes were significantly downregulated in these plants (Fig. S1). These results suggest that Cr acts as a signaling molecule that initiates antioxidant defense mechanisms in grains and that associated genes were involved in Cr detoxification, although they primarily exhibited inhibitory effects. Further, the generation of ROS has the potential to severely harm DNA and interfere with intracellular DNA replication and repair processes. The DEGs of Profile11 from the STEM analysis (Table S4) were enriched in functions related to ribosomal protein complex formation in addition to methylation modification of tRNA and rRNA. Moreover, GO and KEGG enrichment analyses (Fig. 3, Fig. S1) indicated increased expression of genes related to DNA replication and associated complex assembly, including the DNA replication licensing factor-MCM complex that is a component of replication decapping enzymes required for DNA replication initiation and elongation in eukaryotic cells (*Remus et al., 2009*). The upregulation of genes related to MCM complexes suggests that DNA chromatin state is critical for mediating the harmful effects of chromate. KEGG analyses (Fig. 3) revealed that genes involved in mismatch repair and nucleotide excision repair were primarily enriched, while COG analyses (Table S3) revealed that genes related to replication, recombination, and repair were primarily enriched. Most Cr-induced DNA damage occurred during the early stages of Cr stress and increased grain resistance by promoting the expression of genes involved in DNA replication and repair, which are crucial in preventing genomic changes caused by Cr stress.

In addition to the above, plant defense system and phytohormone signaling genes were upregulated in order to resist Cr stress. Further, GO enrichment analysis (Fig. S1) revealed that defense and damage regulation-related genes were considerably upregulated in the Cr_6h and Cr_6d plants. HM stress activates the defense systems of plants to protect themselves from damage. Plants can chelate metal ions as critical defense mechanisms against stress by using specific metal-binding ligands like MT and PC (*Srivastava et al., 2021*). Cr detoxification involves the reduction of Cr (VI) to Cr (III) and the formation of PC-Cr complexes for transport to vesicles for detoxification. In addition, Cr toxicity in plants results in the production of PCs, wherein PC gene expression is induced to mitigate accumulated Cr in the roots and leaves of mustard and other plants (*Shahid et al., 2017*). GO and KEGG analyses (Fig. 3, Fig. S1) revealed that GSH metabolism and glutathione-S-transferase (GST) gene expression were all upregulated. GSTs comprise a multifunctional enzyme family that can bind to GSHs to induce cellular detoxification while also providing tolerance to Cr stress by regulating peroxide reduction and scavenging free radicals (*Seregin & Kozhevnikova, 2023*). Hormone signaling also controls how plants grow and respond to stress. JA-mediated signaling pathways were considerably upregulated in the Cr_6h and Cr_6d plants, with over half of the 132 signaling pathway-related genes being upregulated, including 71 JA, 74 ABA, and 71 IAA. ABA signals through the ethylene response (ETR1) pathway have been shown to inhibit root growth in *Arabidopsis*. HM exposure can induce upregulation of the JA pathway, leading to rapid increases in the JA content in rice and bean plants, while exogenous JA application can significantly reduce the root lengths and fresh weights of *Arabidopsis* seedlings (*Hao et al., 2021*). Thus, Cr stress likely triggers the production of PCs and defensive mechanisms in cereal leaves to protect

against Cr stress, while phytohormones like ABA and JA may control the inhibitory effects of Cr (VI) on the growth of cereal seedlings.

## Resilience of soil microbial communities to Cr stress

The composition of soil bacterial and fungal communities were significantly altered after Cr stress. The dominant phyla of the bacterial communities were Actinobacteriota, Firmicutes and Proteobacteria (Fig. 5), while the dominant genera were *Planifilum*, *Longispora* and *Actinomadura*. Proteobacteria have been shown to be the dominant bacterial taxon in habitats contaminated with As and Cd while also exhibiting high resistances to HMs (*Jiang et al., 2019*). Further, several Proteobacteria have been shown to effectively transform Cr (VI) (*Garavaglia, Cerdeira & Vullo, 2010*), which is frequently used as a biological indicator of Cr pollution. The *Bacillus* genus of Firmicutes, in addition to the *Arthrobacter* and *Streptomyces* genera of Actinobacteriota, exhibits a high resistance to Cr and can also reduce the toxic effects of Cr by decreasing Cr (VI) levels (*Liu et al., 2019a*). However, the abundances of Firmicutes significantly decreased in the Cr_6d soils, likely because the HM concentrations exceeded the tolerance of the bacterial communities, thereby inhibiting their growth. Microbial communities also adapt to soil HM pollution by altering its composition (*Li et al., 2017*). Consequently, Cr enriches more tolerant bacteria while inhibiting the growth of more sensitive bacteria.

The most dominant fungal phyla were the Ascomycota, Basidiomycota and Chytridiomycota, while the most dominant genera were *Chaetomium*, *Gibberella* and *Fusarium* (Fig. 5). Ascomycota have enormous catabolic capacities and adaptability, likely leading to their prevalence and dominance in the fungal communities, while Basidiomycota are widely distributed in agricultural soils and can efficiently transmit HMs. Increased Cr stress time led to significant decreases in the relative abundances of *Aspergillus* and *Penicillium*. Both taxa are widely distributed in soils, are resistant to Cr, and can convert Cr (VI) to Cr (III), with a high capacity for biosorption of Cr (*Dewi, Mumpuni & Yusiana, 2020*). *Penicillium* and *Aspergillus* became less abundant with increased Cr exposure, perhaps related to the greater toxicity of Cr stress to fungi. *Mortierella* and *Gibberella* are more sensitive to Cr levels, and their abundances dramatically decreased in the Cr_6h soils. In addition, *Fusarium* abundances significantly increased. The pathotrophic nutritional strategies of *Fusarium* collect nutrients by destroying plant roots, although they are resistant to Cr stress (*Tedersoo et al., 2015*). Thus, Cr stress prevented the growth of sensitive taxa while enriching the growth of more resilient taxa.

While bacterial community diversity first decreased and then increased with Cr stress exposure time, fungal community diversity primarily only decreased (Fig. 6). The resultant physiological and biochemical toxicity of habitats contaminated with Cr to bacterial and fungi cells (*Hu et al., 2018*) may be the cause of altered microbial community diversity, with changes in community structure and diversity being obvious manifestations. However, bacterial communities are frequently robust in soils contaminated with HMs due to their diverse substrate usage and high metabolic activities, while fungal communities are more vulnerable to HM exposure (*Pasqualetti et al., 2012*). Further, dominant soil bacterial

populations might compete with fungal populations for resources, placing additional stresses on fungal populations.

*Beta* NTI values (Fig. 7) and co-occurrence networks (Fig. 8) revealed that soil bacterial community assembly processes changed from being primarily influenced by deterministic processes to being primarily influenced by stochastic processes with increasing Cr exposure, while stochastic processes remained dominant for fungal communities throughout the treatments. The interplay between deterministic and stochastic processes determines how soil microbial communities are constructed, while the relative importance of the two processes varies depending on the environment in question. Stochastic mechanisms of community formation can influence community assembly to a greater extent than deterministic processes and can produce a wider range of ecological functions. In addition, stochastic processes produce beneficial feedback mechanisms in sustainable agroecosystems, reducing ecological disruptions caused by Cr and thereby preserving ecosystem stability (*Knelman & Nemergut, 2014*). Concomitantly, deterministic processes also drive the development of communities and microbial ecological network features that characterize how microbial communities respond to stress and the connections between related ecological niche activities (*De Vries et al., 2018*). Increased Cr stress exposure led to decreases in the edges, mean degrees, enhanced modularity, and mean path lengths of bacterial networks. In contrast, the mean degree and modularity of the fungal network decreased, while the mean degree and mean path lengths of nodes in the fungal network increased. The bacterial co-occurrence network was larger, exhibited greater associations, and less modularity than the fungal network (Table 2). Nevertheless, both co-occurrence networks exhibited predominately positive correlations, with different taxa enhancing community resistance primarily through inferred mutually beneficial interactions that could enhance positive synchronous feedbacks to perturbations caused by Cr (*Schimel & Schaeffer, 2012*). Increased mean path lengths would serve to decrease the potential responses of microbial communities to Cr stress (*Ma et al., 2020*) and would facilitate the use of soil nutrients by microorganisms. Networks with higher connectivity also generate greater resistance to environmental perturbations. The differences in the underlying mechanisms of the bacterial and fungal communities may stem from the differences in cell sizes between the two groups, since cell sizes are closely related to species growth rates, community assembly mechanisms, and community dispersal characteristics (*Luan et al., 2020*).

## Effects of Cr stress on the abundances of carbon and nitrogen cycling functional genes

In Cr-stressed soils, increased abundances of functional genes related to carbon and nitrogen cycling were observed (Fig. 9), although only AOA-*amoA* and AOB-*amoA* exhibited significant increases in gene abundances ($p < 0.05$). Ammonia oxidation to nitrite is the rate-limiting step in soil nitrification (*Prosser et al., 2020*), with AOA and AOB mediating this process and generating significant amounts of ATP that helps microbial populations survive in soils (*Luan et al., 2020*). The dominance of Actinobacteriota and Acidobacteria in the soils may be related to increased transcription of plant genes related

to amino acid production and absorption as a means of protein detoxification, potentially explaining the increase in the abundances of genes involved in the nitrogen cycle (*Viti et al., 2014*). Cr treatment had little to no effect on the abundances of *pmoA* and *mcrA* genes, possibly due to the absorption of Cr (VI) by plant roots and secretions, thereby alleviating stress on microorganisms. Root secretions at the root interval are generally methanogenic and favor the survival of methanogenic bacteria, while oxygen that is inhibitory to methanogenic bacteria can be transmitted to the atmosphere through plant aeration tissues, thereby enhancing the abundance of methane-oxidizing bacteria (*Wu, Ma & Lu, 2009*). Thus, oxygen excretion may significantly influence the stability of functional gene abundances related to carbon cycling. Overall, soil microbial communities gradually responded to external Cr stress at the genotype and community levels. These results suggest that to sustain high energy costs during organismal growth and metabolism, organisms may accelerate the use and conversion of nitrogen and carbon sources to survive in Cr-stressed habitats.

## CONCLUSIONS

The accumulation of Cr caused significant changes ($p < 0.05$) in the rhizosphere soil and physicochemical properties and biomass indicators of cereal plants, resulting in toxic effects. (1) Millet plant responds to Cr stress by regulating the expression of relevant genes, including those related to photosynthesis, cell division, cell membranes, DNA replication, signal transduction, plant defense mechanisms, and phytohormones. To adapt to the Cr-stressed environment, millet significantly inhibits its own photosynthesis by reducing chlorophyll content, photosynthetic system activity, and affecting structural components such as thylakoids. Simultaneously, it down-regulates the expression of cell wall and microtubule-related components to inhibit the proliferation and differentiation of leaf cells. Furthermore, millet also activates the plant's defense system to overcome toxicity. (2) Soil fungal and bacterial communities adapt to Cr stress by adjusting their compositions, interspecific relationships, and the expression of functional genes. Further, Cr stress may have inhibited the growth of Cr-sensitive species (*Planifilum*, *Gibberella*, *etc.*) while increasing the abundances of resistant taxa (Actinobacteriota and Ascomycota, *etc.*). Moreover, both co-occurrence networks exhibited predominately positive correlations, with different taxa enhancing community resistance primarily through inferred mutually beneficial interactions. Further, the abundances of microbial functional genes showed an overall upward trend. Microorganisms may accelerate the use and conversion of nitrogen and carbon sources to enhance positive synchronous feedbacks to perturbations caused by Cr stress.

## ACKNOWLEDGEMENTS

We are grateful to all the scientists who contribute to the collection of data used in this study. We thank LetPub platform reviewers for their valuable comments.

### Funding

This study was supported by the Science and Technology Innovation Programs of Higher Education Institutions in Shanxi (Grant number 2020L0535), the Construction of Innovation Discipline Cluster Servicing Valley Ecological Governance Industry (Grant number Shanxi "1331 Project"), and the Critical Talent Workstation Project (TYSGJ202201). The funders had no role in study design, data collection and analysis, decision to publish, or preparation of the manuscript.

### Grant Disclosures

The following grant information was disclosed by the authors:
The Science and Technology Innovation Programs of Higher Education Institutions in Shanxi: 2020L0535.
The Construction of Innovation Discipline Cluster Servicing Valley Ecological Governance Industry: 1331 Project.
The Critical Talent Workstation Project: TYSGJ202201.

### Competing Interests

The authors declare there are no competing interests.

### Author Contributions

- Pengyu Zhao conceived and designed the experiments, analyzed the data, authored or reviewed drafts of the article, and approved the final draft.
- Yujing Li performed the experiments, prepared figures and/or tables, and approved the final draft.
- Xue Bai conceived and designed the experiments, performed the experiments, analyzed the data, prepared figures and/or tables, authored or reviewed drafts of the article, and approved the final draft.
- Xiuqing Jing analyzed the data, authored or reviewed drafts of the article, and approved the final draft.
- Dongao Huo performed the experiments, authored or reviewed drafts of the article, and approved the final draft.
- Xiaodong Zhao performed the experiments, prepared figures and/or tables, authored or reviewed drafts of the article, and approved the final draft.
- Yuqin Ding performed the experiments, prepared figures and/or tables, and approved the final draft.
- Yuxuan Shi analyzed the data, prepared figures and/or tables, and approved the final draft.

### Data Availability

The sequences are available at NCBI: PRJNA930506.

## Supplemental Information

Supplemental information for this article can be found online at http://dx.doi.org/10.7717/peerj.17461#supplemental-information.

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
