# Peer review of "Resistance mechanisms of cereal plants and rhizosphere soil microbial communities to chromium stress"

_PeerJ, doi:10.7717/peerj.17461_

## Round 0.1 · original submission · Major Revisions

Dear Dr. Li,

Both reviewers were consistent in their assessment of the work. I agree with their opinion. I would like to ask you for the making of a thorough revision of the work, please take into account all the comments of both reviewers.

·

Basic reporting

The manuscript "Resistance mechanisms of cereal plants and rhizosphere soil microbial communities to chromium stress" proposed to the publication in PeerJ describes morphological, biochemical and molecular consequences in examined plants after the exposition to heavy metal stress (chromium stress). The aim of the work was to investigate the potential mechanisms of plant resistance to chromium and to examine the changes in microbial rhizosphere community, as well as expression of bacterial genes in response to chromium stress. In my opinion manuscript is interesting and is a valuable sorce of knowledge about the impact of chromium on plants and the soil environment.
However, in my opinion manuscript needs to be checked trough profesional Editor, who correct minor linguistic errors and technical inconsistencies in the text. For example: (1) the title of Table 1 does not contain information about chlorophyll and nitrogen content, (2) sometimes there is luck of space or double space, (3) in Table 1 some results are written 1.0±0.1a and some 1.0 ± 0.1a - it needs to be unified. Also, I recommend to prepare list of abbreviation. Moreover, sometimes abbreviations in tabel titles or in Tables are undeveloped.
The manuscript has sufficient background and literature is appropriately referenced.
Please check the labels of the Figures and Tables - they need to describe precisely what Figure/Table shows.
Presented aims of the study were stated in Introduction and achieved by Authors during the research.

Experimental design

Proposed manuscript with obtained findings are comparable with Aim and Scope of the Journal. Plants and microbes response to heavy metal stress is one of the most important issue in novel plant biology. Therefore, obtained original results in the manuscript are important source of knowledge and will be useful for other scientists. However, Authors need to clarify/complete some gaps in the text: (1) in Materials & Methods "Materials and experimental design" there is no plant species name (only the name of variety). Plant species name should be indicated in Latin and English., (2) in the same section light conditions used for plant cultivation are not enough described - please characterise the source of light, light quality and quantity, (3) in Measurements and analysis "Plant physiological indicators and biomass measurements" the method for chlorophyl and nitrogen measurement is not clear. Please describe briefly how methods work. (4) Line: 242: Please clarify what kind of post hock test were used.

Validity of the findings

Presented manuscript is original. Authors describe effect on chromium on various levels of environment (plant/microbes in soil). Some of the phenomenons showed in article were known before, but whole work gives quite wide insight into the chromium effect on living organisms.
I believe that results are characteristic by high quality (Authors used recognized scientific methods and statistical tools).
Unfortunately, conclusion section needs to be rewritten. Conclusions proposed by the authors are now a repetition of the results. Based on the results, Authors should draw conclusions based on known biochemical/molecular processes that are disturbed by chromium.

Reviewer 2 ·

Basic reporting

In some expressions small changes should be done, e.g.,
- line 189 “Soil …analysis” or “DNA … analysis of soil community”?
- line 385 “Soil bacterial and fungal community α and β diversity” or “α and β diversity of soil bacterial and fungal community”? Literature references are up to date.
There are some problems with the tables. I cannot see Table S1 (line 224), Table S3 (line 328), Table S4 (line 347). There is no Table S2 mentioned at all, thus I cannot check these parts in the manuscript. What is the “MCM complex” (line 296)?
Description of all figure panels is missing, e.g. Figs 1 and 6 - no description of panels A-D, Fig. 4 - no description of panels A and B.
I also recommend using the abbreviation “Cr” instead of “chromium” in all figure labels (Figs. 2, 3, 5 and 6).
Fig. 2 – Please, check if the third column (Cr 6h&Cr 6d) is correctly presented. The first set of data is marked red, and the next is blue, which is opposite to the other two columns. Are colours changed or data mixed? Because of these discrepancies, I am unable to evaluate this data.
Fig. 3 – panel CK&Cr_6h up – please, add number to black point/dot and panel CK&Cr_6d down – please, improve the font/size.
Fig. 5 – What one, two and three asterix mean? It should be explained. I propose to widen all four panels to be able to present letters indicating statistical significance because you are writing about more than the first three phyla or genra.
Fig. 6 – What does a black dot in panels A and B mean? It should be clarified. Please, add the median value as a red point/dot. In panel A improve the X axis label from “insex” to “index”.
Fig. 7 – What does a black dot in panels A and B mean? It should be clarified.
Tables 1 and 2 – Please, reorganize the first row and write “Chromium stress” in the middle of two columns concerning Cr.
Fig. S1 – Please, check and correct the font size in panel Cr_6h&Cr_6d up.

Figs and Tabs – The statistical convention is that the highest value is denoted by letter “a”, a slightly lower value by “b”, and so on. It should be standardised.

The term 'resistance' often accompanies the term 'tolerance'. It would be useful to briefly, in 1-2 sentences, distinguish between these two defensive strategies in order to also convince of the validity of the choice of title of the manuscript (line 1), the goal of the study (line 114), and subchapter in the Discussion section (line 462).

Results section
Please, rewrite the whole passage concerning “Growth analysis” (lines 261-264). Write about statistically significant changes. It is not 22.40 but 22.47%. Decreased or increased compared to … - you have presented three groups of CK. This is very unclear.
Check also line 292 for MF category.
Lines 311-312 – delete space bar after “cereals” and before “.”.
Lines 364, 365 – “Actinobacterota” or “Actinobacteriota”?
Lines 365-384 need considerable rewriting. It is unclear what “increased and then decreased” (e.g., lines 366, 367, 368-369, 372, 377) means precisely. The focus should be on statistically different results, without mixing statistical significance with high or low values without statistical significance. Please, check that your statements are in accordance with Fig. 5 (e.g., line 372, 377, 381-382). The statement "while ..." (lines 372-373) cannot be evaluated because there are no letters that mean statistical significance.
Lines 389-390 – the statement “significantly … trend” is unclear and needs clarification (see comments for lines 365-384).
Line 397 – needs explanation; which figure/s confirm this statement.
Line 400 – add “(Fig. 6C)” before “and” and “(Fig. 6D)” before “were”.
Line 418 – move “(Table 2)” after “characteristics” or delete it because this table was cited at the end of the sentence.
Line 428 – It would be good to change “different … differed”.
Lines 453-457 – The statements in these two sentences need to be checked. Please, mention only those data that are statistically significant.
Lines 457-459 – Again, this statement is not based on statistical significance (Fig. 9), and does not seem to apply to pmoA (46.19%) and mcrA (33.01%).

Discussion section
1. It is unclear whether the statement is based on the literature or on the Authors’ own research. I suggest citing appropriate figures or tables when discussing the Authors’ data. This will make the section easier to follow. Examples are mentioned below.
An appropriate citation (literature/figure/table) should be provided in lines 466-469, 476, 484, 488, 492, 500, 509, 511, 515, 524, 525, 530, etc.
Please, read the whole discussion carefully and fill in the missing citations in the text.
2. Improvement in the “Resistance …” section (lines 463-551) is necessary as it is now more about plant response than strictly resistance.

Experimental design

1. Introduction – I suggest a clearer explanation of the gap in the knowledge. The phrase “have not been systematically investigated” (line 107) requires a detailed explanation based on the literature.
2. Please, complete full latin and English name of the plant – line 123.
3. “a unique substrate” (line 125) require specific characteristic.
4. “a selection of premium … seeds” – please, clarify what does it mean?
5. An explanation is required as to why in Table 1 we can see three control groups labelled 0h, 6h and 6d despite the fact that CK appears throughout the manuscript with no temporal distinction.
6. It is incomprehensible why Table 1 consists of two tables, with the second table being a repetition of the data from Table 1 in the case of Cr, and a mixture of data in the case of CK:
- with CK 0h (the first two data),
- data similar to that of CK 0h (the next two data),
- a value not matching any value from CK (fifth value)
- with CK 6h (sixth value).
Please, reorganise it appropriately and present only one table.

Other remarks concerning Materials & Methods section are mentioned below.
7. Why is “Leng et al” cited in line 141? Is the sentence in lines 142-144 based on your experiments or based on the literature? Please, clarify.
8. Line 151 – “leaves” instead of “samples” will be more precise.
9. Line 159 – “fresh weight of the whole plant” is more precise. Please, complete this information.
10. Line 186 – use abbreviation “CK” – it was explained before.

Validity of the findings

The conclusions should be improved. There are no “resistance” issues appropriately supported by the results.

Additional comments

1. Abstract should be verified. More specific results should be presented. The sentence in lines 29-31 is unclear. Line 30 – p value can be deleted here. “Functional responses” (line 38) should be briefly explained.
2. Introduction – Some changes are necessary.
- To make the text more concise, sentences from lines 73-82 should be moved to line 52 (before “Agricultural …”). Next, organize this Cr(III) and Cr(VI) issue after transfer.
- Omit some repetitions and try to concise lines 86-87 with 103-104.
3. References require unification. Some examples are stated below.
- Lines 674, 735 (space bar after “:”) vs. line 676 (lack of space bar after “:”).
- Lines 705, 717 – “arabidopsis” – use first capital letter, please.
- Line 707 – comma should be after “W J” but not “.”
- Line 710 – Unnecessary space bar after “J”.
- Line 737 – “please, use a full name of “Earth Environ”.
- Line 739 – please, complete the data.
- Line 741 – “cadimium”???
- Lines 753, 755, 756, 767, 768, 791-793, 801-802, 809, 815, 816 –capital letters in the middle of the title words are used in minority, so they require standardisation.
- Line 819 – doubled space bar.
- Latin names are usually written in italics – not only in references but also in the text of the manuscript (alpha – line 245, beta – line 406, via – line 465).

---

## Round 0.2 · Minor Revisions

I thank the authors for improving the manuscript, however, please note that the reviewers still found minor flaws that should be corrected.

·

Basic reporting

No comments.

Experimental design

No comments.

Validity of the findings

No comments.

Additional comments

After the first review, the authors improved the text of the manuscript. They also made suggested corrections to the tables and conclusions.
The only thing that was not prepared by the authors is a list of abbreviations, which I think would make the manuscript easier to read. However, I leave the decision to prepare it to the authors.

Reviewer 2 ·

Basic reporting

Although that general language is quite correct, some small changes should be made. The examples are marked in the pdf file of the manuscript.
I am unable to evaluate changes made by Authors in the figure and table descriptions because I cannot find them. It includes most of the comments from Basic reporting section, point 3 of Authors response (except for 2 comments concerning Figs 2 and 3 and the second and third comments regarding Fig. 6).
Unfortunately, in Authors' answers to the reviwer's comments, there are many mistakes in providing line numbering, e.g. in Basic reporting point 1 Authors declared adjustments in lines 186 and 381 but they are in lines 188-9 and 382, and in Experimental design point 10 - these are not lines 146, 159 and 184, but 147, 160 and 186, etc.

Results section
Point 1 - based on lines from 365 to 390 (reviewer's previous comments). This part of the text should be improved because not all statements are in accordance with the figure. Also sentence from lines 452-453 should be improved. Sentences or parts of the sentences are marked in the pdf file of the manuscript.
Point 2 - Please, improve English and specify - lines 392-393.

Experimental design

Only full English name of the plant studied is missing.

Validity of the findings

No comment.

Additional comments

No comment.

Annotated reviews are not available for download in order to protect the identity of reviewers who chose to remain anonymous.

---

## Round 0.3 · Minor Revisions

Dear Authors,

Please make the minor changes that the Reviewer suggested.

Reviewer 2 ·

Basic reporting

No comment.

Experimental design

No comment.

Validity of the findings

No comment.

Additional comments

I have attached some changes in the text of the manuscript.

Annotated reviews are not available for download in order to protect the identity of reviewers who chose to remain anonymous.

---

## Round 0.4 · Minor Revisions

The Section Editor noted a methodological error that must be addressed.

The methods state that TPM normalized counts were used for differential gene expression analysis in DESEQ2, but DESEQ2 requires raw counts. See https://www.bioconductor.org/packages/release/bioc/vignettes/DESeq2/inst/doc/DESeq2.html#why-un-normalized-counts

Please recalculate correctly and check that it does not affect your conclusions.

---

## Round 0.5 · accepted · Accept

Dear authors, thank you for improving the methodology.